# Using Multimodal Deep Neural Networks to Disentangle Language from Visual Aesthetic Experience

## Abstract

When we experience a visual stimulus as beautiful, how much of that experience derives from perceptual computations we cannot describe versus conceptual knowledge we can readily translate into natural language? Disentangling perception from language in visually-evoked affective and aesthetic experiences through behavioral paradigms or neuroimaging is often empirically intractable. Here, we circumnavigate this challenge by using linear decoding over the learned representations of unimodal vision, unimodal language, and multimodal (language-aligned) deep neural network (DNN) models to predict human beauty ratings of naturalistic images. We find that unimodal vision models (e.g. SimCLR) account for the vast majority of explainable variance in these ratings. Language-aligned vision models (e.g. SLIP) yield small gains relative to unimodal vision. Unimodal language models (e.g. GPT2) conditioned on visual embeddings to generate captions (via CLIPCap) yield no further gains. Caption embeddings alone yield less accurate predictions than image and caption embeddings combined (concatenated). Taken together, these results suggest that whatever words we may eventually find to describe our experience of beauty, the ineffable computations of feedforward perception may provide sufficient foundation for that experience.

## 1 Background

Imagine a beautiful sunset; then imagine how you might describe it to your friends. What words might you use to capture what made this particular sunset beautiful, compared to other sunsets that you've seen before? How confident would you be that those words accurately convey the "feeling" of that experience? How much would your friends experience that beauty through your words?

Aesthetic experience (the experience of beauty) is a universal phenomenon without a universal definition. Centuries of debate, from antiquity onwards, have asked why we experience beauty, and where it comes from (Ross, 1951; Tatarkiewicz, 2006; Reber, 2012; Chatterjee, 2014; Palmer et al., 2013; Menninghaus et al., 2019; Graham, 2019; Skov and Nadal, 2020; Redies et al., 2020; Isik and Vessel, 2021; Vessel, 2022). A central theme in these debates is the notion of ineffability: the extent to which our experience of beauty can be adequately described in natural language (Kant, 1987). Given the inherent subjectivity of affective self-report, researchers have in many cases attempted to better operationalize ineffability by localizing or attributing our experience of beauty to various points along an axis, which at one end conceptualizes aesthetic experience as the product of a highly encapsulated process that is inaccessible to language and at the other assumes beauty is the product of conscious, deliberative, *verbalizable* thought (Vessel and Rubin, 2010; Schepman et al., 2015; Shimamura and Shimamura, 2012; Redies, 2015; Brielmann and Pelli, 2017).

These debates are challenging and difficult to arbitrate with behavior (i.e. empirical aesthetics) or neuroimaging (i.e. neuroaesthetics). In this work, we suggest that one potential route for moving this debate forward is with the use of computational models (i.e. computational aesthetics) in the form of deep neural networks (Brielmann and Dayan, 2022). Deep neural network models trained on canonical computer vision and natural language processing tasks allow us to systematically control the kinds of computations and information processing

mechanisms a given system can use to make inferences about aesthetic stimuli. Here, we use a linear decoding method to assess how well we can predict human ratings of beauty for a diverse set of naturalistic images from the features of unimodal and multimodal deep neural network models never trained explicitly on predictions of beauty. Our main goal in this is to better understand the relationship between representation learning and aesthetic experience, and how various task modalities modulate that relationship.

## 2  METHODS

Our main source of human ratings in these experiments is the OASIS dataset (Kurdi et al., 2017), a set of 900 images curated to span a 7-point scale of arousal and valence ratings, and to which ratings of aesthetics were later added (Brielmann and Pelli, 2019). Each image comes with a rating that is the average of 100 to 110 human raters. To predict these group-average affect ratings, we use cross-validated regularized (linear) regression over features extracted from (pretrained) deep neural network models, none of which receive any prior training on aesthetic targets. To compute these regressions, we proceed layer by layer through each network, extracting the features and decoding the aesthetic ratings from these features in a procedure designed to mimic standard methods (e.g. MVPA (Haxby, 2012)) for (supervised) linear decoding from brain recordings. That is to say, we use each feature map to predict how subjects will rate an image, then correlate those predicted ratings with the actual ratings provided by the participants. The higher the correlation, the more information about aesthetics is available in a given feature map, with no more than a linear regression necessary to convert network activity into an aesthetic prediction. See Figure 2A and Appendix A.2 for details.

The logic here is one of representational sufficiency: If the predictions of our feature regressions are accurate, it suggests that whatever the underlying computations producing aesthetics in the human brain may be, they need not be any more sophisticated than a single affine transformation of the kinds of representation produced by the feedforward, hierarchical operations of a deep neural network. In this analysis, we use this logic to probe what kinds of deep net representations are sufficient for predicting aesthetics, and better triangulate the computational pressures (i.e. tasks) that produce them.

In this particular analysis, the pressures of interest are primarily at the level of the training data (i.e. image pixels or tokenized words) – which define a given model's modality. "Unimodal vision" models in this schematic are models that learn solely from images via self-supervision. (Category-supervision, in the form of explicit training on one-hot category labels, introduces a linguistic confound). "Unimodal language" models in this schematic are models that learn solely from tokenized text, again via self supervision (masked or next word prediction). "Multimodal models" are models that learn from vision and alike, usually, but not exclusively through self-supervision. By the logic of representational sufficiency, comparing these models in controlled experiments allows us to more directly isolate the kinds of information – visual, linguistic, and mixed – that are sufficient for the prediction of human beauty judgments.

## 3  RESULTS

All scores reported in these results are in units of 'explainable variance explained': the squared Pearson correlation coefficient between predicted and actual ratings divided by the squared Spearman-Brown splithalf reliability of the ratings across subjects (the 'noise ceiling'). Given the quantity of subjects underlying the average, the noise ceiling for this data is extremely high at $r_{Pearson} = 0.988$ [0.984, 0.991]. Unless otherwise noted, we report the score of a model's (cross-validated) maximally predictive layer as that model's overall score.

**Unimodal Vision Models** In line with previous work, we first show that pure unimodal vision models, in the form of contrastive (self-supervised) image models, are capable of predicting up to 75% of the explainable variance in the group-average beauty ratings. From a sample of 18 contrastive learning models that learn only over augmented image instances

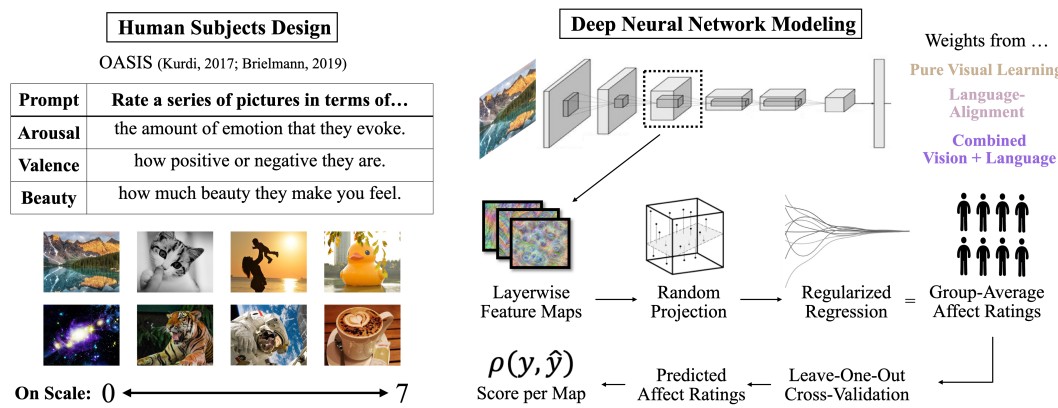

Figure 1: Schematic of our feature regression pipeline for decoding affective information from deep net responses. Our target in these experiments are group-average beauty ratings, which we predict by extracting image features from a candidate deep neural network model, (optionally) reducing their dimensionality, then employing them as predictors in a cross-validated ridge regression with the group-average beauty ratings as output. This method gives us a beauty decoding score per layer per candidate model.

(e.g. Dino, SimCLR, SWaV), the average explained variance is 0.607 [0.566, 0.641]. The most predictive model, a RegNet64 trained using the SEER pretraining technique (Goyal et al., 2021) explains 74.6% of explainable variance. While trained using roughly a billion images, this model's representations are learned *without* any form of symbolic (i.e. linguistic) training targets. This means that models trained on *images alone* can account for the majority of explainable variance in human beauty ratings.

**Multimodal Vision Models** The CLIP models (Radford et al., 2021) are a series of models trained on the task of linguistic alignment: given an image and a caption paired with that image, the model encodes both in an equidimensional latent space, computes the cosine similarity between them, then (during training) back-propagates any similarity less than 1 as a loss term. The representations of the visual encoder are thus directly shaped by language. OpenAI's CLIP models (S/16, B/32, L/14, et cetera) all show small, but significant gains over the best-performing unimodal image model (RegNet64-SEER), with 80.5% to 87% of explainable variance explained.

The problem, however, with comparing the CLIP model directly to other models is that CLIP is trained on a proprietary dataset of 400 million image-text pairs not yet available to the public. To address this discrepancy, we use the SLIP models (Mu et al., 2021) – a series of Vision Transformers (Small [ViT-S], Base [ViT-B], & Large [ViT-L]), all trained on the YFCC15M dataset (15 million image-text pairs), but only on 1 of 3 tasks: pure SimCLR-style self-supervision; pure CLIP-style language alignment; or the eponymous SLIP – a combination of self-supervision and language alignment. The SLIP models allow us to control for the influence of language, holding architecture and dataset constant. (A schematic of this controlled modeling procedure involving the SLIP models may be found in Figure 3A).

The pattern of results across the SLIP models (Figure 3B) (and in particular the comparison between SimCLR and SLIP) suggests *adding language* to purely visual learning does indeed increase the downstream predictive accuracy of aesthetic ratings. Specifically, while pure CLIP-training shows discrepant gains over pure SIMCLR-training across the 3 vision transformer sizes (performing slightly better in ViT-S and ViT-B, and slightly worse in ViT-L), SLIP-training outperforms its pure SimCLR counterpart across all 3 transformer sizes by a significant, at least midsize margin. A bootstrapping analysis using 1000 resamples of the human subject pool (averaging across model size) shows the difference between SimCLR and CLIP to be nonsignificant, with a bootstrapped mean of 0.0098 [-0.027, 0.041] ($p = 0.67$),

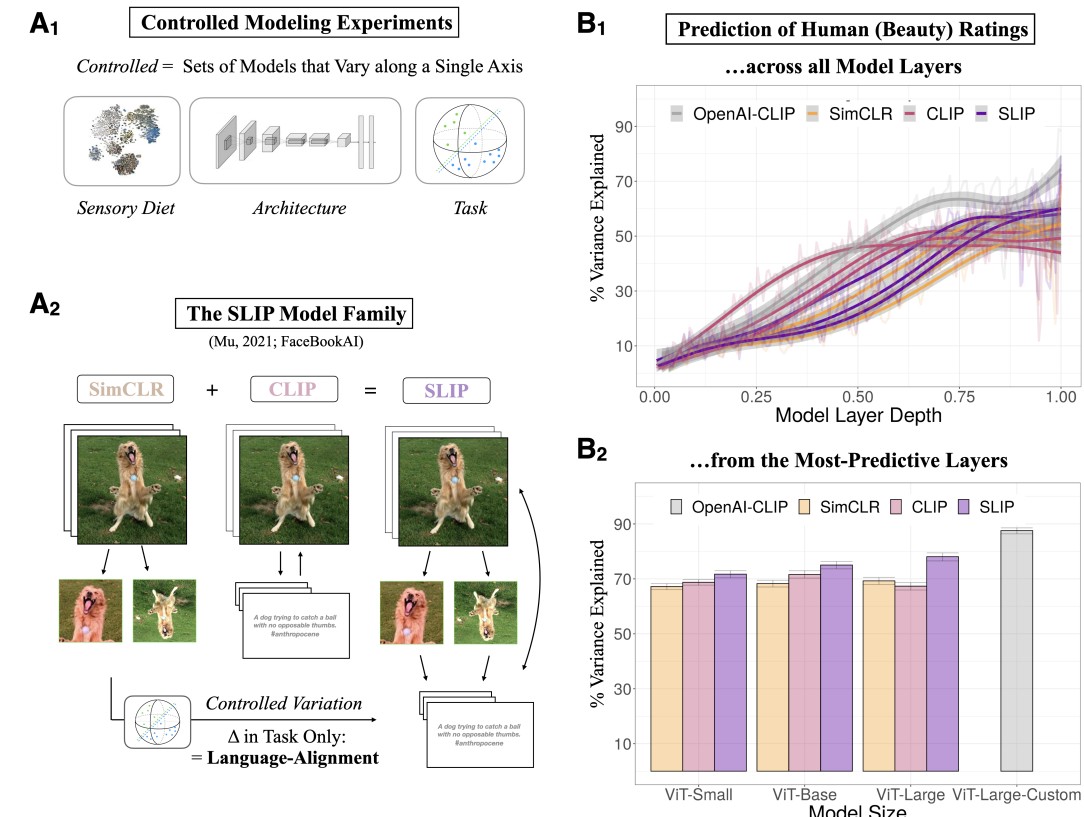

Figure 2: **A** Schematic of our controlled modeling experiment using the SLIP model family (Mu et al., 2021). "Controlled" in this case refers to the isolation of singular axes of interest across distinct sets of model that vary exclusively along these axes (with other possible variations held constant). In SLIP, both the training dataset (YFCC15M) and architecture (ViT-[S,B,L]) are held constant across 3 variants of model (SimCLR, CLIP, and SLIP). The difference between SimCLR and SLIP (a combination of SimCLR's visual augmentation regime with CLIP's language alignment in a unified contrastive learning pipeline) are a direct empirical instantiation of variation in the presence or absence of training provided by language. **B** Results from our feature regression pipeline as applied to SimCLR (a unimodal vision model), CLIP (a language-aligned model) and SLIP (a model that combines unimdal vision training and language alignment) – holding dataset and architecture constant. **B₁** In the top plot, we see results across layers (the semitransparent jagged lines are individual layer scores; the curves are the output of a generalized additive smoother across layers; the SLIP models each have 3 variants: ViT-[Small, Base, Large]). The takeaway here is that for all models, predictive accuracy is generally higher in deeper layers (with the final embedding layer often the highest). **B₂** In the bottom plot, we see the results from the maximally predictive layers of each model. Error bars are 95% confidence intervals across 1000 bootstrap resamples of the human subject pool. The takeaway here is that adding language alignment (without taking away unimodal vision training) in the form of the SLIP objective does significantly increase downstream readout of aesthetic information.

while the difference between SimCLR and SLIP is significant, with a bootstrapped mean of 0.067 [0.037, 0.096] ($p$ ¡ 0.001).

**Language Models via Captions** Adding language to visual representations by way of CLIP-style alignment (in concert with contrastive visual augmentation regimes) does seem to facilitate better downstream prediction of aesthetic ratings. But what exactly is language doing here? Is it really just adding to the visual representation or is it changing that

representation in some fundamental way? To assess this, we opted to test the outputs of a unimodal language model *conditioned* on CLIP's visual encoder using our feature regression pipeline. This required first converting the visual embedding generated by CLIP into an embedding suitable for a language model. For this, we used an adapter module called CLIP-Cap (Mokady et al., 2021). CLIP-Cap is a closed-loop system that employs a small multilayer perceptron (MLP) or transformer model to project the visual embedding from a CLIP model to a token embedding – called a 'prefix embedding' – that can be used by GPT2 (Radford et al., 2019) to generate a natural language caption.

For this experiment (summarized with detail in Figure 3), we use CLIP-Cap's MLP method of projection, which defaults to a prefix embedding length of 10 and uses CLIP-ViT-B/32 as its visual backbone. In the same way we decode aesthetics from features evoked by images in visual models, here we decode aesthetics from features evoked by the 'embeddings' (for prefix and caption alike) in the language model: that is to say, layer by layer, and using the same regression method. We find first and foremost that while the projected visual prefix embedding preserves all the information necessary to decode aesthetics as accurately as in the CLIP visual encoder, the hierarchical language processing of GPT2 facilitates no additional decoding. (The accuracy of CLIP's visual encoder is 84.8% [83.2%, 85.6%] explainable variance explained; the accuracy of GPT2 operating over the prefix embedding never exceeds 85.3%.).

In this case, then, the features evoked across the language model do not seem to be adding information – though neither do they seem to be losing it. This invites the question of whether language alone might be sufficient for capturing the variance explained with the prefix embedding. To test this, we took the most probable caption generated from the GPT2 model for each prefix embedding, and passed that caption back through the model with the prefix removed. While we found these captions were unable to account for the full 85% of explainable variance explained by the vision-conditioned prefix embeddings, we found them capable of explaining a nontrivial 38.6% [37.2, 40.1] of explainable variance in aesthetic ratings. Count-vectorized embeddings of these same captions explain only 19.4% [18.6, 20.1] of the explainable variance – suggesting the predictive power of these language features is not attributable to single-word concepts (or confounds) alone.

**Better Captions, Better Language Models** Our experiment with the translation (machine to machine) of vision into language via end-to-end captioning does leave open the possibility that better language models and better (more accurate, or more descriptive) machine-generated captions could close the gap on the variance explained by visual models per se. Even state-of-the-art captioning models make consistent, common-sense errors no human would make in describing an image (Wang et al., 2022a). What does this mean for our current experiment with automated captioning?

One point to consider is that we are not necessarily interested in the accuracy of the caption per se, but the extent to which that caption reflects the information content available in the visual embeddings of CLIP, which themselves may not accurately reflect category-level or more generally semantic content. The issue then is not whether CLIP-Cap (or other systems that interpret CLIP's visual embeddings in service of caption generation, such as Cho et al. (2022) provides accurate human-legible captions, but whether those captions reflect a coherent summary function of CLIP's visual embeddings. This is admittedly difficult to measure, but because CLIP-Cap and similar models are gradient-based, we can say definitively, at least, that the resultant captions are literal functions of CLIP's vision. Another potential issue with the use of machine-generated captions specifically in this pipeline are the large language models we use to transform those captions into embeddings appropriate for our feature regression pipeline. CLIP-Cap uses as its language transformer a standard (midsize) GPT2 model. Language models are known to be far more accurate with scale (Kaplan et al., 2020). Could other language models (in conjunction with better captions) facilitate greater decoding accuracy?

While by no means an exhaustive experiment, we explored this question by expanding our caption-based decoding paradigm to two other sets of captions and two other large language models. For captions, we considered CLIP-Caption-Reward (Cho et al., 2022) (another CLIP-based caption-generation algorithm that uses CLIP similarity as a reward function)

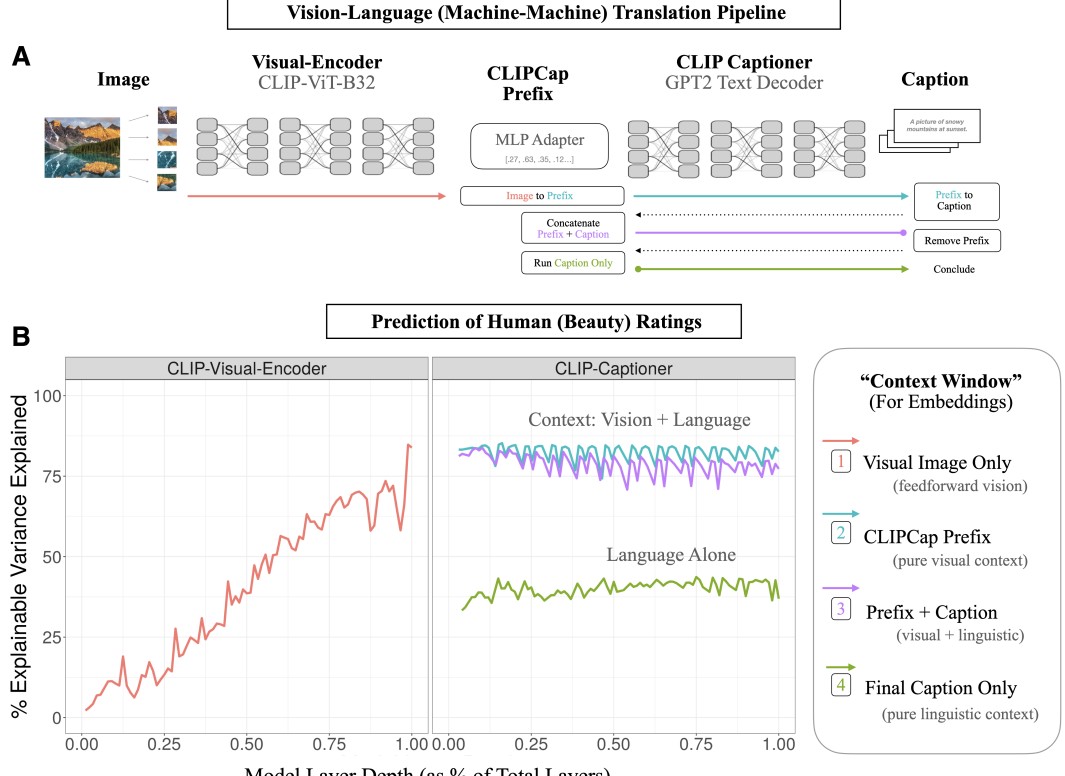

Figure 3: **A** Schematic of our experiment using CLIPCap (Mokady et al., 2021) to translate the visual embeddings of CLIP into natural language by way of a GPT2 text decoder: The process begins with the embedding of an image (red line) into the latent space of a CLIP-ViT-B32 model. These embeddings contain only feedforward visual information. CLIP's latent visual embedding is then piped into GPT2 by way of CLIPCap's MLP adapter, and in the first pass through GPT (blue line), the only context available to GPT2 for next token generation is the visual information instantiated in CLIPCap's "prefix" tokens. Once a caption is produced, we concatenate (purple line) this caption with the original visual prefix and pipe it once again through GPT2 to extract embeddings that instantiate both the original visual information in the prefix, as well as any added information instantiated in the caption. Finally, we remove the visual prefix from the caption, and extract the GPT2 embeddings for the generated caption alone, effectively extracting the pure linguistic context provided by this caption. **B** Results of the CLIPCap translation experiment: The red line in the facet on the left are the scores across the layers of the CLIP visual encoder used to generate an image 'prefix' embedding that is subsequently passed to GPT2 for captioning. The line in blue in the facet on the right is the predictive power of that prefix embedding as it is processed across the layers of GPT2. In other words, this blue line tracks the potential of GPT2 to facilitate better aesthetic decoding by extracting further information from the visual prefix. The line in green is the predictive power of the generated caption passed back through GPT2 *without* the prefix embedding. This line tracks how well (machine-generated, image-conditioned) language alone might predict aesthetic ratings. The line in purple is the predictive power of the generated caption passed back through GPT2 *with* the prefix embedding. This line tracks whether visual embeddings and image-conditioned language together might outperform either one alone. The difference between the blue line and the green line represents the difference in predictive power between CLIP's visual features and GPT2's linguistic features – the difference, in other words, between language-aligned perception and language alone. This gap is substantial. The negative slope on the purple line seems to be an artifact of the feature regression overfitting to the embedding complexity added by the caption. Each line in this plot may be thought of as instantiating a form of "context window" – a term used in natural language processing to describe one information provides precedent for any given "next token" prediction in the language-generating process.

and GIT (Generative Image-to-Text Transformer) (Wang et al., 2022a). For language models, we considered GPT2-XL (the larger version of the GPT2 used by CLIPCap for caption generation) and the All-MPNet-Base-V2 variant of S-BERT (Reimers and Gurevych, 2019) (the largest thereof). While no single caption and model combination exceeds 58% of explainable explained variance (compared to the visual encoder's 82%), the best combination (SBERT-over-GIT captions), improves nearly 20% over the baseline we test in the main results (GPT2-over-CLIPCap) at 38.5%. This latter caption-model combination notably does not involve CLIP, which makes it irrelevant as a method of interpreting the CLIP visual encoder's predictive accuracy, but it does suggest one potential route forward for assessing the impact of language on aesthetic judgment. A more detailed summary of these experiment results may be found in Table 1 below.

Table 1: Model Results with Confidence Intervals and Scores

| Model | Score | | |
| --- | --- | --- | --- |
| | Mean | Lower CI | Upper CI |
| CLIP-ViT-B/32-over-Images | 0.827 | 0.818 | 0.835 |
| GPT2-over-CLIPCap | 0.386 | 0.372 | 0.401 |
| GPT2-over-CLIPReward | 0.464 | 0.447 | 0.481 |
| GPT2-over-GIT | 0.424 | 0.407 | 0.440 |
| GPT2XL-over-CLIPCap | 0.385 | 0.368 | 0.401 |
| GPT2XL-over-CLIPReward | 0.452 | 0.435 | 0.469 |
| GPT2XL-over-GIT | 0.478 | 0.457 | 0.496 |
| SBERT-over-CLIPCap | 0.516 | 0.505 | 0.527 |
| SBERT-over-CLIPReward | 0.548 | 0.536 | 0.560 |
| **SBERT-over-GIT** | 0.599 | 0.586 | 0.610 |

( Colored row corresponds to the reference (vision) model.)

## 4 CONCLUSION

Aesthetic experience is no single phenomenon, but a pluralistic combination of multiple different factors: our sensory and social ecologies, our bodies, our idiosyncratic developmental trajectories, our beliefs, and our perceptions (Biederman and Vessel, 2006; Shimamura and Shimamura, 2012; Redies, 2015; Germine et al., 2015). An overarching goal of this and similar works is in some sense to approximate what percentage of aesthetic experience may be attributable to certain kinds of computational processes (Brielmann and Pelli, 2017; Redies et al., 2020). Here, we show that while perceptual processes in the form of feedforward, hierarchical, subsymbolic visual feature extraction are so far the best predictors of how people on average will rate the aesthetics of naturalistic image stimuli, language (alignment) may play a statistically meaningful role in shaping these representations. Furthermore, it seems that whatever the nature of the visual semantics that undergird the successful prediction of aesthetic responses in multimodal models like CLIP, at least a nontrivial portion of these semantics may be translated to machine-generated natural language descriptions. Aesthetic ineffability in this sense may be less of a binary (effable or ineffable) and more of a gradient. The difference between the predictive power of an image in visual feature space and its description in natural language space could serve as a direct quantification of this gap.

Of course, this exact same point makes clear a few inherent limitations to some of the methods we've used here: simply put, not all image descriptions are made equal. Just as an expert orator may be more capable of evoking emotion with language than a novice, so too might certain descriptions communicate aesthetic value more effectively than others – even without explicitly affective qualifiers. (Our experiment with better caption models certainly suggests as much). Exposition of key details or interactions in a scene might be essential to communicating its aesthetic quality. To the extent that this is true adds immense complexity to the endeavor of disentangling vision from language, but the use of machine

vision and language models does potentially allow us to pursue this disentanglement in ways that weren't necessarily available to experimentalists before.

An important caveat to the use of these models in empirical pipelines, however, is that it requires a great deal of conservatism that may (at first glance) seem somewhat out of step with the current zeitgeist of large-scale generative artificial intelligence (e.g. the development of powerful, and increasingly multimodal, LLM-based chatbots such as ChatGPT) (c.f. Zador, 2019; Bowers et al., 2022), and the near-daily production of state-of-the-art models whose latent embeddings may subserve highly accurate predictions of a wide range of phenomena in behavior and brains alike (e.g. (Wang et al., 2022b; Haskins et al., 2023)) – including aesthetics (Hentschel et al., 2022; Xu et al., 2023). This conservatism need not *necessarily* be applied to the further development of these models (whose applied competence suffices as evidence of progress), but it should be applied to any inferences we make about the computations of the human mind based on the computational internals of these models. We believe that such inferences can in most cases be made more rigorously on the basis of controlled model rearing (c.f. (Wood et al., 2020)) like the ones allowed for by distinct "sets" of models like the SLIP family.

In terms of future work for this particular application of multimodal DNNs to aesthetics research, one immediate priority to assess the extent to which methods like consensus-based caption-scoring (Vedantam et al., 2015) could be used to reconcile divergent natural language descriptions of the same stimulus into a single representation – something that might allow us to supplement our machine-generated captions with crowdsourced human captions. Aggregating multiple natural language descriptions into a single coherent embedding might also be the key to closing the distance between visual representations and natural language descriptions that match these representations in terms of their downstream predictive power. Other, less proximate work should reconsider what it would mean for an affective experience (like the experience of beauty) to be communicated effectively between one agent and another, and whether this kind of communication has implications for learning.

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

# A    Appendix

## A.1    Code, Data, & Compute Specifications

The OASIS dataset is publicly available available under a Creative Commons License at the following URL: https://osf.io/6pnd7/ All code will be made available upon publication. All experiments were run on a single Linux machine with 8 RTX3090 GPUs and 756GB of RAM. Most computations were CPU intensive and GPU use could be avoided entirely.

## A.2    Method Details: Feature Regression

Our feature regression pipeline consists of 4 distinct phases: feature extraction; dimensionality reduction; ridge regression; cross-validation and scoring.

**Feature Extraction** We consider feature extraction from 'every layer' to mean the sampling of network activity generated after each distinct computational suboperation in a deep neural network model. This means, for example, that we consider a convolution and the nonlinearity that follows it as two distinct operations that produce two distinct feature spaces, both of which we consider candidates for decoding. If a layer returns a tensor with multiple components (such as a convolutional layer) we first flatten the tensor to a single component, such that the layer represents any given image as a feature vector. The layer thus represents a dataset of $n$ images as an array $\mathbf{F} \in \mathbb{R}^{n \times D}$, where $D$ is the dimensions of the feature vector.

**Sparse Random Projection** For some deep-net layers $D$ is very large, and as such performing ridge regression directly on $\mathbf{F}$ is prohibitively expensive, with at best linear complexity with $D$, $\mathcal{O}(n^2 D)$ (Hastie and Tibshirani, 2004). Fortunately it follows from the Johnson-Lindenstrauss lemma (Johnson, 1984; Dasgupta and Gupta, 2003) that $\mathbf{F}$ can be projected down to a low-dimensional embedding $\mathbf{P} \in \mathbb{R}^{n \times p}$ that preserves pair-wise distances of points in $\mathbf{F}$ with errors bounded by a factor $\epsilon$. If $u$ and $v$ are any two feature vectors from $\mathbf{F}$, and $u_p$ and $v_p$ are the low-dimensional projected vectors, then;

$$(1 - \epsilon)||u - v||^2 < ||u_p - v_p||^2 < (1 + \epsilon)||u - v||^2 \tag{1}$$

1 holds provided that $p \geq \frac{4 \ln(n)}{\epsilon^2/2 - \epsilon^3/3}$ (Achlioptas, 2001). With $n = 900$ for our dataset, to preserve distances with a distortion factor of $\epsilon = .1$ requires $\geq 5830$ dimensions. Thus we chose to project $\mathbf{F}$ to $\mathbf{P} \in \mathbb{R}^{n \times 5830}$ in instances where $D >> 5830$. To find the mapping from $\mathbf{F}$ to $\mathbf{P}$ we used *sparse random projections* following Li et al. (2006). The authors show a $\mathbf{P}$ satisfying 1 can be found by $\mathbf{P} = \mathbf{F}\mathbf{R}$, where $\mathbf{R}$ is a sparse, $n \times p$ matrix, with i.i.d elements

$$r_{ji} = \begin{cases} \sqrt{\dfrac{\sqrt{D}}{p}} & \text{with prob. } \dfrac{1}{2\sqrt{D}} \\[2ex] 0 & \text{with prob. } 1 - \dfrac{1}{\sqrt{D}} \\[2ex] -\sqrt{\dfrac{\sqrt{D}}{p}} & \text{with prob. } \dfrac{1}{2\sqrt{D}} \end{cases} \tag{2}$$

**Ridge Regression with LOOCV** We used regularized (ridge) regression to predict the average human ratings of images, $\mathbf{Y}$, from their associated (dimensionality-reduced) deep net features, $\mathbf{P}$. As our goal was not to identify a particular regression model for later use, but rather get a best estimate for the linear read-out of beauty scores from deepnet feature spaces, we utilized all the data at our disposal with a leave-one-out (generalized) cross-validation procedure. For every image in our dataset ($\forall i \in \{1 \dots 900\}$) we fit the coefficients $\boldsymbol{\beta}_i$ of a regression model on the remaining data, such that $\mathbf{Y}_{-i} = \mathbf{P}_{-i}\hat{\beta}_i + \epsilon$ with minimal $\|\epsilon\|$ (error). Ridge regression penalizes large $\|\hat{\beta}\|$ proportional to a hyper-parameter $\lambda$, which is useful to prevent overfitting when regressors are high-dimensional (as with $\mathbf{P}$). We first standardized $\mathbf{Y}$ and the columns of $\mathbf{P}$ to have a mean of 0 and standard deviation of 1. Let $\mathbf{P}_{-i}$ and $\mathbf{Y}_{-i}$ denote $\mathbf{P}$ and $\mathbf{Y}$ with row $i$ missing, then each $\hat{\beta}_i$ is calculated by;

$$\hat{\beta}_i = \left(\mathbf{P}'_{-i}\mathbf{P}_{-i} + \lambda I_p\right)^{-1} \mathbf{P}'_{-i}\mathbf{Y}_{-i} \tag{3}$$

Each $\hat{\beta}_i$ is then used to predict the beauty rating from the deepnet feature projection of each left out image;

$$\hat{y}_i = \mathbf{P}_i\hat{\beta}_i, \quad \hat{\mathbf{Y}} = \{\hat{y}_i\}_{i=1}^{900} \tag{4}$$

The hyper-parameter $\lambda$ we set at $1e4$, a value we determined using a logarithmic grid search over $1e\text{-}1$ - $1e6$ on an AlexNet model that we subsequently exclude from the main analysis. $\lambda = 1e4$ yielded the smallest cross-validated error ($\|\mathbf{Y} - \hat{\mathbf{Y}}\|$) when averaging across layers. We used the *RidgeCV* function from (Pedregosa et al., 2011) to implement this cross-validated ridge regression, as its matrix algebraic implementation identifies each $\hat{\beta}_i$ in parallel, resulting in significant speedups (Rifkin and Lippert, 2007).

**Scoring** In this analysis, we *score* each deepnet layer by computing the Pearson correlation coefficient between its predicted ratings, $\hat{\mathbf{Y}}$, and the actual group-average affect ratings from the human subjects, $\mathbf{Y}$. To convert this Pearson correlation coefficient into a score that represents the percentage of explainable variance explained, we divide the square of this coefficient by the square of the Spearman-Brown split-half reliability that constitutes the noise ceiling.

Note that previous empirical work suggests the sparse random projection step in this pipeline is largely optional and can, without substantial decrease in accuracy, be eliminated in favor of directly using the full-size, flattened feature maps in the regression.

