# OpenReview forum: "Using Multimodal Deep Neural Networks to Disentangle Language from Visual Aesthetic Experience"
_ICLR.cc/2025/Conference — Submitted to ICLR 2025_

### Official Review · Reviewer_WpaE · 2024-10-30

**Soundness:** 3
**Presentation:** 2
**Contribution:** 2
**Rating:** 5
**Confidence:** 3

**Summary:**

This paper looks at how perceptual and linguistic factors can contribute to human judgements of aesthetics. It does so by examining correlations between feature representations of vision, language, and vision and language models to human judgements of beauty on the OASIS dataset, a collection of 900 images. The paper finds that unimodal contrastive image models are capable of predicting up to 75% of the explainable variance, while multimodal vision and language models can explain up to 87% of the explainable variance. In general, vision and language models provide a small (but significant) improvement in explained variance, suggesting that language-image pre-training can help models to further understand human concepts of 'aesthetics'.

**Strengths:**

This paper has several notable strengths:
- The idea of trying to disentangle the impacts of language and vision features in the perception of beauty is fairly interesting from a theoretical point of view: is it vision features alone that contribute to our perception of beauty, or is language also somewhat impactful? The question is novel, and thought provoking, and as a reader, I am quite interested in the outcome (despite it's actual usefulness in any practical application).
- Reiterating the novelty - as far as I am aware there is no existing work that investigates this question, and the approach provides an interesting way of extracting insights into human perception from trained neural networks.
- The models compared are fairly comprehensive, and there is a strong effort made to control for variables across the models (for example, the use of the SLIP architecture). Further, the approach is quite rigorous in avoiding many issues with the small dataset (such as using cross-validated ridge-regression with a regularized linear decoding pipeline to avoid overfitting).
- The paper provides some interesting and concrete empirical evidence that language may help our understanding of aesthetics.

**Weaknesses:**

While the paper is interesting, it has several weaknesses (among the other questions, discussed in the questions section):
- One of the key issues is that the findings are restricted to the OASIS dataset, which is only 900 samples, collected by MTurk users residing in the United States. This suggests a fairly biased view of "aesthetics", for something that the paper claims is "a pluralistic combination of multiple different factors: our sensory and social ecologies, our bodies, our idiosyncratic developmental trajectories, our beliefs, and our perceptions." Given these claims, I wonder if these results would generalize at all to a more diverse set of ratings, and the paper would benefit significantly from a more diverse evaluation across more than one aesthetic dataset.  Similarly, the paper focuses on group-average aesthetic ratings, which overlooks individual correlated variations in aesthetic performance.
- Many of the models investigated in this paper are quite old (GPT-2, CLIP, CLIP-Cap, etc.), and the paper would benefit significantly from using updated models (Dino V-2, SigLIP, Llava, etc.), which might help to solve some of the issues which the paper identifies with poor modeling. For example, using Llava could help to answer the question on L264 without resorting to the more obtuse experiments performed in Table 1. Another option could be experimenting with human ground truth captions, and extracting LLM features from those ground truth captions - which might provide additional insight not tied to caption model quality).
- The paper does not contain any baseline approach for understanding the %Explained Variance measure. While the language-alone baseline (presented in Fig 3.B) does help to center the results, it would be helpful to have several additional points of reference (such as random feature sets, untrained resnets, HoG/Gabor features, etc.).
- The paper only looks at a linear decoding approach to assess the predictive power of the model embeddings. While this does simplify the interpretation of the method, there's (in my opinion) likely to be a much more complex relationship between model features and aesthetic judgement, as aesthetics is unlikely to be entirely a type 1 system (which is more likely to be linearly correlated with feature vectors). Experimenting with some non-linear methods (SVMs or simple neural networks) could also be quite interesting.
- The paper primarily focuses on pearson correlation to assess performance, which, while useful, can have some limitations, particularly in higher dimensional feature spaces (or in noisy feature spaces). It would be good to look at some secondary measures beyond just pearson correlation.
- This paper is lacking significant detail on the methods, which is relegated to Appendix A.2. Indeed, without reading this appendix, it is very difficult to understand exactly what experiments were performed, and how the approach computes feature correlations. While the method may be somewhat tangential to the presented results, I believe that it is necessary to include at least some of these key details in the main paper.
- While this isn't exactly a weakness, I'm not entirely certain that this paper has any applications beyond theoretical curiosity. While the findings might have broader implications across multimodal modeling, and the approach could be applied for more effective studies, it might be good for the paper to include some more concrete applications of either the methods, or the results/conclusions (should they end up being true).

**Questions:**

- Instead of using the SLIP architecture, could you use a model set such as OpenCLIP, which retains the architectural capabilities, but has the required open-source requirements?
- It seems like there is a lot of inter-layer variance in Figure 2, B.1 - Is there any insight into why this might be happening? Or things that this inter-layer variance can tell us about the impact of model architecture on aesthetics perception?
- In Figure 2, B.2 - it seems like there is little impact of model size on %variance explained - is there any intuition as to why this might be?
- Is there any intuition as to why RegNet outperforms ViT style models (such as Dino/SimCLR)?
- It seems like the key difference that the paper identifies between OpenAI's CLIP model and the other CLIP models is training data -- are there data scale effects here that might impact the results (i.e. does training with more data allow models to express more of the variance in aesthetic ratings)?
- The justification in L259 for if CLIP-cap reflects a summary of CLIP's visual embeddings is quite weak, and is (technically) incorrect, as while the **distribution** of captions is a gradient-based function of the CLIP visual embeddings, the captions themselves are a sample from this distribution, and thus reflect only a (possibly quite small) subset of the potential information in the CLIP embeddings themselves. Using a more powerful language model could help to compensate for this, but this depends a lot on the decoding methods of the captioning model - for example, using beam-decoding/maximum likelihood decoding from a strong model will never generate the full distribution of potential captions (only those from the maximum likelihood mode). It would be good for the paper to both (1) clarify this, and (2), potentially consider averaging or otherwise aggregating across multiple sampled captions from strong models to improve the results.
- How does the inter-annotator agreement for each of the OASIS samples impact the ability to extract aesthetics from the models? It seems like the correlation should be far higher for samples with high annotator agreement, than those with lower agreements.

More minor comments:
- It would be really nice from a presentation perspective to have a table of methods and explained variances. It's quite challenging to compare the results of the models, since a lot of the actual explained variance numbers are buried in the text of the paper.
- In Figure 2, A.1 - is the sensory diet the training dataset? Is there something more to this?
- Some of the key results in the paper are fairly obvious (and perhaps vacuously true): (1) Vision+Language models explain more variance than vision models alone (which is likely to be true, given that additional data is used in the form of language content) and (2) Language representations of visual data explain less variance than visual features (which must be true, as the captions contain a subset of the information contained in the image data). While not all results should be surprising, I think that it might have been more exciting to take an approach which has the potential to generate more remarkable results.
- There's no limitations section (in the main paper, or appendix), where some of the issues with the work could be discussed
- Several parts of the paper claim that certain models are state of the art, when they are not.

---

### Official Review · Reviewer_R6wp · 2024-11-01

**Soundness:** 3
**Presentation:** 2
**Contribution:** 3
**Rating:** 5
**Confidence:** 4

**Summary:**

This paper investigates the relationship between visual perception and language in aesthetic experience using deep neural networks. By comparing unimodal vision models, multimodal vision-language models, and language models processing generated captions, the study aims to disentangle how much of our experience of beauty stems from pure visual processing versus linguistic conceptualization. Using the OASIS dataset of 900 images with human beauty ratings, the researchers employed linear decoding over different neural network representations to predict aesthetic judgments.

The results reveal that unimodal vision models alone can explain up to 75% of explainable variance in beauty ratings, while language-aligned models (like CLIP and SLIP) show modest but significant improvements, reaching up to 87% accuracy. In contrast, caption-based language models only explained about 39% of the variance. These findings suggest that while language plays a meaningful role in shaping aesthetic representations, the foundation of aesthetic experience may primarily rely on perceptual processing rather than linguistic conceptualization. The study quantifies the gap between visual and linguistic representations in aesthetic judgment, suggesting that beauty exists on a gradient of expressibility rather than being purely describable or indescribable in language.

**Strengths:**

1) The paper leverages unimodal and multimodal deep neural networks to predict aesthetic experiences, which provides a novel approach to disentangling perceptual (visual) from conceptual (language-based) components in aesthetic judgment. It leverages models such as SimCLR, SLIP, and CLIP in a controlled framework to advance the understanding of how different types of embeddings contribute to the complex experience of beauty.
2) The paper provides a theoretical understanding of aesthetic ineffability by quantifying the predictive gap between visual and language-conditioned models. The authors’ use of linear regression and variance-explained metrics allows them to empirically explore how well each model captures the "ineffable" aspects of beauty, thus providing a quantitative foundation for studying the ineffable in aesthetics.
3) The paper explores vision-only, language-only, and language-aligned vision models to offer a perspective on how each type of model contributes differently to aesthetic experience, something that hasn’t been extensively explored.

**Weaknesses:**

1) The paper’s reliance on the OASIS dataset limits generalization, as OASIS primarily represents a homogeneous sample.  Perhaps, the inclusion of datasets like AVA or BAM! datasets, which include user-generated ratings and varied aesthetic preferences, can help to capture more diverse perspectives.

2) There is no ablation study on the usefulness of their approach across diverse cultural groups. It would be useful to conduct experiments for comparing aesthetic predictions across culturally specific subsets or diverse contexts. This could reveal whether the model's predictive accuracy holds across diverse user groups or if fine-tuning these demographic-specific data improves accuracy.

3) CLIPCap and GIT image captioning models used in this paper tends to produce straightforward, often literal captions that may not fully capture the high-level semantics or emotional resonance often associated with aesthetic experiences. This can limit the model's ability to align visual embeddings with complex, subjective aspects of beauty, as captions generated by these models may lack descriptive richness and specificity. CLIPCap is restricted by the training data, thus it lacks the complexity needed for rich aesthetic descriptions. Therefore, the captions generated may underperform in capturing subjective qualities that influence aesthetic appeal, such as mood or emotional tone, which are vital for a nuanced aesthetic prediction. For instance, the paper (3a) referenced below, suggests that richer descriptions help bridge the gap between human aesthetic judgment and machine representations - highlighting that language's contribution is more valuable when it goes beyond the literal.

3a) Yang, Xu, Jiawei Peng, Zihua Wang, Haiyang Xu, Qinghao Ye, Chenliang Li, Songfang Huang, Fei Huang, Zhangzikang Li, and Yu Zhang. "Transforming visual scene graphs to image captions." arXiv preprint arXiv:2305.02177 (2023).

4) The paper does not leverage attention mechanisms, which are often used to highlight parts of an image that are critical to aesthetic judgment. Studies like MMLQ referenced below have shown that targeted, learnable queries or attention maps can improve aesthetic predictions by allowing the model to focus on specific visual features or regions. Without an attention mechanism, the model may treat all parts of the image equally, potentially diluting its focus on aesthetically significant regions.  While CLIPCap and GIT use attention within their transformer layers, they lack targeted, aesthetic-focused attention mechanisms that prioritize image regions based on aesthetic relevance.

4a) Z. Xiong, Y. Zhang, Z. Shen, P. Ren and H. Yu, "Multi-modal Learnable Queries for Image Aesthetics Assessment," 2024 IEEE International Conference on Multimedia and Expo (ICME), Niagara Falls, ON, Canada, 2024, pp. 1-6, doi: 10.1109/ICME57554.2024.10687472.

**Questions:**

1) A deeper analysis of how language alignment enriches visual embeddings could be useful. Could the authors provide insights into which specific linguistic features (e.g., descriptive adjectives, emotional terms) contribute most to the performance boost?

2) The results highlight that certain layers within the model offer higher aesthetic prediction accuracy. Could the authors clarify which layer types or depths (e.g., shallow vs. deep) contribute most, and why?

3) The study’s reliance on the OASIS dataset might limit generalizability, especially to diverse cultural contexts or artistic styles. Could the authors discuss any potential bias introduced by this dataset and suggest how it could be addressed? Performing experiments on more diverse aesthetic datasets can help to corroborate the potential of the proposed approach.

4) The results are primarily quantitative, focusing on the variance explained. Have the authors compared the model’s outputs with human qualitative judgments, especially for ambiguous or highly subjective aesthetics for a small subset of the data?

---

### Official Review · Reviewer_xKm1 · 2024-11-02

**Soundness:** 3
**Presentation:** 2
**Contribution:** 2
**Rating:** 3
**Confidence:** 4

**Summary:**

This paper explores the disentanglement of language from visual aesthetic experience using multimodal deep neural networks (DNNs). It investigates the extent to which perceptual computations and conceptual knowledge contribute to the experience of beauty. The authors employ linear decoding over representations from unimodal and multimodal DNNs to predict human beauty ratings of images. Key findings suggest that unimodal vision models explain most of the variance in beauty ratings, with minimal gains from language-aligned models and no additional gains from unimodal language models. The paper contributes to computational aesthetics by examining the sufficiency of different DNN representations for predicting aesthetic judgments.

**Strengths:**

* The paper tackles a complex and intriguing question at the intersection of aesthetics, perception, and language, providing novel insights into the role of language in visual aesthetics.
* The paper presents a comprehensive analysis with various models, including unimodal and multimodal DNNs, providing a detailed view of how different types of information contribute to aesthetic judgments.

**Weaknesses:**

* The authors should consider testing their method on larger and more diverse datasets to ensure the results can be extrapolated to different types of visual stimuli beyond the OASIS dataset (small dataset, 900 images).
* The method proposed in the paper is relatively simple, which may limit its applicability and robustness in more complex scenarios.
* The authors need to do more ablation studies on the contribution of individual components in multimodal models to support the conclusions.
* The paper has some formatting issues and needs a more comprehensive discussion of related work.

**Questions:**

* How might the results differ with different image datasets or aesthetic criteria?
* What are the authors' thoughts on the potential cultural biases in the aesthetic ratings, and how might this affect the universality of their conclusions?

I look forward to an active discussion with the authors during the rebuttal phase and will revise my score accordingly.

**Details Of Ethics Concerns:**

The paper don't have ethical concerns.

---

### Official Review · Reviewer_upqx · 2024-11-04

**Soundness:** 1
**Presentation:** 2
**Contribution:** 1
**Rating:** 3
**Confidence:** 4

**Summary:**

This paper investigates how much of our experience of visual beauty comes from perceptual computations versus conceptual knowledge expressed through language. The researchers used various deep neural network (DNN) models to predict human beauty ratings of natural images, comparing Unimodal vision models (trained only on images), Multimodal vision models (trained on both images and language)
and Language models processing machine-generated image captions, and found that combined visual and linguistic features performed better than either alone.

**Strengths:**

1. Propose the use multimodal vision language model for solving the aesthetics perception problem for a better explanation, this is a pretty nice experiment
2. The figure is well-drawn and expresses the main idea of the each experiments
3. The experiments have clear isolation of language effects from visual processing

**Weaknesses:**

1. This paper is not a well-written paper and does not have a clear structure.
    - It misses the related works section, which is important and necessary.
    - The citation format is incorrect and inconsistent with the ICLR official formats
    - Figure's captions are excessively long and thus of a bad presentation style
    - The number of page is not well-controlled
2. The experiments design and introduction is unclear. The metric used for evaluation is "explainable variance explained", described in pure text (line 98 - line 100) instead of formal representations. There are no descriptions in the main context about how the authors actually map the feature map into score and why they choose this method.
3. The experiments are only conducted on OASIS dataset with only 900 images, which can decrease the faithfulness of the paper.

**Questions:**

1. How is the metric computed, can you describe them in formal language? Better with some examples.
2. Could you add related works to the paper?
3. Have you tried using multimodal LLM like GPT-4V, GPT-4o, LLaVA, etc, to conduct the understanding of aesthetics? Simple prompt engineering to generate a score from 0 to 7 shall be pretty simple to do and better aligns the state-of-art multimodal research.

---

### Meta-Review · Area_Chair_HYcq · 2024-12-18

**Metareview:**

This paper investigates how much how much of our experience of visual aesthetic is driven by perceptual computations versus conceptual knowledge expressed through language. the author explores how DNN predict human aesthetic ratings on the OASIS dataset.
The main findings indicate that unimodal vision models shows a modest boost (up to 87%) and language only model explain much less variance. it concludes that aesthetic rating rely primarily on visual cues but still benefit from linguistic input.
This paper further investigate the interplay between visual perception and language in shaping aesthetic, a relatively unexplored area. and shows empirical evidence that vision is essential but language adds some incremental value. but this finding relies on a small dataset less than 1000 images limiting generalizability, and most of models are outdated, it should explore the latest multimodal LLMs
Also Missing deeper ablation studies limits the value of this work, and explainable variance is described mostly in text.
There was no rebuttal submitted by the authors, and the reviewers showed a consistent consensus toward rejection.

**Additional Comments On Reviewer Discussion:**

There was no discussion and new changes from author,
Instead, I leave additional summary comments on the points raised by the reviewers.
- This paper applies new multimodal models to see if language alignment yield better results
- and investigates aesthetic judgements on novel dataset
- analyzing linguistic features to understand which concept matter most
- potential cross-cultural bias and variations in judgement

Overall, the paper would be more convincing with larger, diverse data and updated models

---

### Decision · Program_Chairs · 2025-01-22

Reject